

# Geoscience Communication: A Content Analysis of Practice in British Columbia, Canada Using Science Communication Models

Courtney C. Onstad[1], Eileen van der Flier-Keller[1]

[1]Department of Earth Sciences, Simon Fraser University, Burnaby, V5A 1S6, Canada

*Correspondence to*: Courtney C. Onstad (courtney_onstad@sfu.ca)

**Abstract.** Geoscience communication, an emerging discipline within the geosciences, faces a scarcity of theoretical grounding despite abundant practical perspectives. This paper addresses this gap by investigating the application of science communication models (deficit, dialogue, participatory) in geoscience communication, specifically in British Columbia, Canada. The overarching aim is to determine if the 'deficit to dialogue' shift often discussed in science communication

literature is reflected in geoscience communication practice. Using a content analysis approach, data was collected from publicly accessible websites to qualify and quantify *how* (activities) and *why* (objectives) geoscience communication practitioners communicate. The activities and objectives were coded based on terms associated with each model that closely aligned with those described by Metcalfe (2019a,b). Findings reveal a persistence of the deficit model in practice (76% for objectives, 61% for activities) with limited adoption of dialogue and participatory approaches. This suggests a discrepancy

between theoretical advancements in science communication and their application in geoscience contexts. The study highlights disparities in the use of communication models across target audiences, regions, and venues. While communication with K-12 audiences utilizes dialogue-based approaches, participatory activities are underrepresented, particularly in regions with high population densities (e.g. Lowermainland/Sea-to-Sky: 0% participatory) and areas where geoscience intersects with public interests (e.g. Northern B.C.: 3% participatory). By shedding light on the current landscape of geoscience communication in

British Columbia, this research informs future endeavours in theory development and practice improvement within the broader field of science communication. However, it also acknowledges the need for localized studies to capture the diverse contexts of science communication practices worldwide.

## 1   Introduction

Geoscience communication has gained recognition as a distinct discipline within science communication, yet there remains a
notable imbalance between practical and theoretical approaches. While practical perspectives offer valuable insights into practitioners' real-world challenges, the absence of robust theoretical frameworks in the geoscience communication literature poses significant concerns (Brossard & Lewenstein, 2010; Nisbet & Markowitz, 2016; Salmon et al., 2017). The lack of theory-informed literature impedes research and undermines the field's credibility, leading to potential misconceptions and limiting the depth, nuance, and applicability of findings to effective practice.



This paper demonstrates the importance of integrating science communication theory into geoscience communication practice. The field of science communication spans diverse contexts and research trends (Dudo & Besley, 2016; Schiele et al., 2012), from citizen science projects to analyzing misinformation during the Covid-19 pandemic (Halliwell et al., 2021, Rzymski, 2021). One enduring and prominent research trend focuses on science communication models, which provide frameworks for understanding how science communicators engage with their audiences (Macq et al., 2020; Metcalfe, 2019b; Nisbet &

Markowitz, 2016). Trench & Bucchi (2010, p.2) note "The near-20 years of discussion of models of science communication – since the naming of the 'deficit model' – is the most solid thread of theoretical work in this field". These models, including the deficit, dialogue, and participatory models, offer insights into the diverse approaches used by science communicators, considering various contextual factors (socio-economic, cultural, historical, and geopolitical). A gradual shift from the deficit to dialogue and participatory models has been observed in science communication rhetoric, reflecting a broader narrative in

the field (Reincke et al., 2020). This research seeks to explore whether this shift is evident in geoscience communication practice. By conducting an environmental scan focused on British Columbia, Canada, we aim to assess the prevalence of deficit, dialogue, and participatory model communication in current geoscience communication efforts. These efforts will be analyzed to determine the alignment between stated objectives and the activities used to achieve them. Identifying alignments or misalignments will provide valuable insights into the consistency, effectiveness, and credibility of messaging, as well as

highlight gaps in current practice. Additionally, the geographic locations and venues where geoscience communication occurs will be examined to assess the prevalence of the various models. British Columbia's unique geoscientific context, its diverse population centres (both rural and urban) and varied venues, offers rich opportunities for geoscience communication.

## 2 Literature Review

### 2.1 Geoscience Communication

The term "geoscience communication" has gained increasing traction within the field of geoscience and Illingworth et al. (2018) underscore its aim to raise awareness of and stimulate discourse on geoscience topics. For this discussion, we will adopt the broadest interpretation, referring to it as communicating geoscience information to non-specialist audiences. Such communication encompasses a wide array of initiatives, including Geoparks, geoscience museums, informal education programs, outreach initiatives, geoheritage projects, and more.

The role of geoscience in addressing society's most pressing challenges is increasingly acknowledged (Stewart & Lewis, 2017; Stewart & Nield, 2013). For instance, some have emphasized geoscientists' role in supporting the United Nations Sustainable Development Goals, which encompass areas such as sustainable development, food security, education, energy, and land use (Franks et al., 2022; Gill, 2017; United Nations, 2015). In light of these significant issues, enhancing geo-literacy through geoscience communication is becoming ever more crucial (Stewart & Gill, 2017). In Canada, this importance was underscored



in the Pan-Canadian Geoscience Strategy, which highlighted increasing geo-literacy as a key recommendation (National Geological Surveys Committee of Canada, 2022). Moreover, discussions surrounding declining enrollment in geoscience departments in Canada have also emphasized the importance of geo-literacy (Council of Chairs of Canadian Earth Science Departments, 2022).

Although calls for improved geoscience communication with the Canadian public date back several decades (e.g. Carleton,
1976), scholarly attention to Canadian geoscience communication remains limited, primarily focusing on broad forms of practice. In designed settings such as public and university museums, efforts to communicate geoscience are evident through permanent exhibits, while the expansion of geotourism, particularly with the establishment of UNESCO Geoparks (Canadian Commission For UNESCO, n.d.), has further bolstered engagement with geoscience (Blackwood, 2009; Royal Ontario Museum, 2021; Additionally, programs aimed at science learning, including outreach activities by university departments and
informal education organizations, as well as ongoing support for educators, contribute to the dissemination of geoscience knowledge and skills (Bank et al., 2009; Dillon & Lipkewich, 2002; Onstad, 2021; Van der Flier-Keller, 2011; Van der Flier-Keller et al., 2011).

## 2.2 Science Communication

The definition of science communication has undergone various interpretations as its study and practice have progressed
(Bucchi & Trench, 2021; Burns et al., 2003). The ongoing debate surrounding the definition of science communication remains relevant today. For instance, Bucchi & Trench (2021) characterize it as "the social conversation around science", emphasizing the interactive nature of communication and encompassing all societal discourse on scientific matters. Besley & Dudo (2022) define it as "communication conducted within the context of scientific issues", a definition adopted in this paper. Furthermore, to narrow the focus of this study, science communication is distinguished from both scholarly communications (occurring
between experts; De Silva & Vance, 2017) and formal science education (aimed at instructional purposes; National Research Council, 2009). However, we acknowledge considerable overlap among these disciplines (e.g. Baram-Tsabari & Osborne, 2015). Practitioners, those who facilitate science communication, are as varied as the field itself (Roedema, 2022). They include scientists, academics, institutions, individuals, and not-for-profits, among others. Target audiences in science communication are individuals or groups engaged with scientific discourse (Ridgway, 2020). While terms like "the public"
are commonly employed, they oversimplify the diverse and intricate nature of audiences (Mohr et al., 2013; Warner, 2002). Target audiences for science communication encompass a spectrum, ranging from broad to specialized, passive to active, local to geographically dispersed, and individualized to collective (Schäfer & Metag, 2021).

Over the years, typologies, frameworks, and models have been proposed to understand the multifaceted nature of science communication activities. For instance, Del Carmen Sànchez-Mora (2016) presents a typology encompassing a broad spectrum
of audiences, facilitators, objectives, activities, and evaluation approaches. Typologies of this nature underscore the intricate



considerations involved in communicating science, highlighting that certain channels may be more effective in conveying messages or engaging specific audiences (Wilson et al., 2017). Media platforms such as television, newsletters, and radio, as well as web-based mediums like social media and podcasts, are frequently utilized for science communication purposes (Huber et al., 2019; Nisbet & Scheufele, 2009; Schäfer et al., 2018). Expanding upon the frameworks used to discuss science

communication, one notable conceptualization is the science communication models, which delineate practitioners' objectives, activities, tactics, and the level of stakeholder and practitioner engagement (Besley & Dudo, 2022; Metcalfe, 2019b; Nisbet & Markowitz, 2016). These models (deficit, dialogue, and participatory) have evolved within diverse contexts, serving varied objectives, and catering to diverse audiences (Dudo & Besley, 2016; Horst, 2012).

The deficit model, often associated with early forms of science communication, aimed to address what was perceived as the

public's lack of knowledge about science (Irwin & Wynne, 1996; Nisbet & Scheufele, 2009; Wynne, 1988). Within this model, publics are perceived as misinformed consequently leading to distrust in the credibility of scientific endeavours (Bucchi & Trench, 2016). Scholars initially believed that rectifying this deficit could be achieved through the one-way transmission of scientific knowledge from experts to the public (Gross, 1994; Irwin, 2006; Nisbet & Scheufele, 2009). However, subsequent research has debunked the idea that mere information provision could effectively change attitudes toward science, revealing

the multifaceted nature of attitude formation influenced by factors such as belief systems, interpersonal interactions, and existing knowledge (Bucchi & Trench, 2008). In contrast to the top-down approach of the deficit model, the dialogue model advocates for a two-way conversation between scientists and the public (Irwin, 2006). Central to this model is the acknowledgment and incorporation of public concerns, opinions, and knowledge into the discourse (Metcalfe, 2019b). The participatory model, in theory, emphasizes a relatively equal standing between the public and scientists (Metcalfe et al., 2022).

Unlike the deficit and dialogue models, the participatory model envisions a process where decision-making power and knowledge are shared between scientists, policymakers, and the public (Schrögel & Kolleck, 2019).

Jenni Metcalfe's work on science communication models has been instrumental in bridging theoretical concepts with practical applications. Her insights, particularly from Metcalfe (2019a), which delineate the objectives and activities associated with each model, serve as a foundational reference for the content analysis in this study. While the science communication models

serve as explicit frameworks for this research, they offer just one lens through which to analyze and interpret science communication practices. Many scholars have proposed continuums to capture the fluid and dynamic nature of these models, recognizing that boundaries between them are often porous and subject to change (Trench, 2008). Embracing these continuums provides a more nuanced understanding of how science communication operates in real-world contexts (Metcalfe, 2019b). Nonetheless, this paper will focus on discussing the models to inform the quantitative and qualitative analyses conducted in

this research.



## 3 Methods

This research aims to qualify and quantify geoscience communication in British Columbia employing a content analysis methodology. Data collection involved the systematic retrieval of information from publicly accessible websites via Google search, with data then organized into a database which included: names of practitioners, target audiences, venues, and other pertinent factors for 146 organizations (also referred to as sampling units). A subset of this data (n=81), specifically from practitioners maintaining public websites, underwent content analysis. Referencing established science communication models (Bauer et al., 2007; Bucchi & Trench, 2008; Metcalfe, 2019a,b), the content extracted from geoscience communication practitioners' online platforms was subject to systematic coding. Qualitative data was also compiled during the content analysis to provide evidence of what practice entails relative to the science communication models.

### 3.1 Database Compilation

Initial data collection involved the identification of relevant terms and keywords associated with the research topic. Keywords such as "geoscience", "Earth science", "geological", "communication", "outreach", "informal education", "museums", and "engagement" along with geographic modifiers such as "British Columbia" or "Vancouver Island" for example, were entered into Google search from April 2022 to March 2024. Only practitioners linked to publicly available websites were considered for inclusion, thereby excluding social media accounts, a common platform for science communication (Huber et al., 2019; Wilson et al., 2016), policy engagements, and other relevant avenues. This criterion was necessary for the content analysis since practitioners' objectives and activities are often explicitly stated on their websites. However, it is worth noting that content analysis of geoscience communication through social media or alternative channels remains feasible, albeit beyond the scope of this study.

The database encompasses various information regarding the target audiences, the resources offered, the primary venue, and the primary geographic location of services offered, among other variables not of focus to this study. Venues were classified based on definitions from existing literature, and natural breaks within the data. Some categories were self-evident, such as museums, which were further classified into history museums and science museums based on distinct objectives. Natural physical sites, as identified by Spector et al. (2012), encompassed locations such as bodies of water where citizen science programs often occur or areas utilized for field trips by rock and mineral enthusiast groups. Parks, including GeoParks and national/provincial parks, were considered distinct from natural physical sites due to their managed nature and human-enhanced elements. K-12 schools and universities were identified as venues for programs for science learning. Online platforms were used as a venue for organizations whose services were available through publicly-accessible websites. It was noted that individual practitioners often operated in multiple venues, and for analytical purposes, a "primary venue" was assigned based on where the majority of their activities were conducted. This process is acknowledged to be subjective and constitutes a limitation of the study.



The incorporation of primary geographic location of services offered and primary venues addresses crucial inquiries regarding accessibility. To achieve this, the locations were classified based on a regional map of British Columbia, which was chosen for its relatively small number of geographic areas (n=7) and its ability to distinguish between geographically distinct and

population-distinct regions. Organizations located outside of British Columbia were included only if they offered online resources accessible to all, excluding those tied exclusively to other provincial or territorial curricula, as such resources would not be relevant to British Columbia's curriculum.

## 3.2 Content Analysis

This study adopts a "problem-driven analysis" approach to content analysis, as outlined by Krippendorff (2018). Content

analysis was conducted on a subset of data extracted from the database. As mentioned previously, only geoscience communication practitioners with publicly accessible websites were included in the analysis. Additionally, geo-art was excluded from the data subset due to the lack of available information regarding the artists' objectives which readily translate to science communication models. As noted by other scholars, sciart may be considered a distinct model of science communication (Bucchi & Trench, 2021). Typically, content analysis involves examining a subset of data drawn from a larger

population (Krippendorff, 2018). However, to obtain the most comprehensive understanding of geoscience communication practices in British Columbia, it was determined to analyze the entirety of the database population, with the exceptions outlined above.

A deductive approach was employed in this analysis due to the strong theoretical foundation underlying the content. Specifically, the analysis was guided by a central research question: To what extent does geoscience communication practice

in British Columbia demonstrate a transition from deficit-oriented approaches to dialogue and participatory models? This question delineates the contextual framework pertinent to the analysis, focusing on the science communication models discerned from practitioners' activities and objectives. By scrutinizing the language used by practitioners to articulate their objectives and activities, and subsequently classifying them according to established science communication models, we can assess the prevalence of deficit, dialogue, and participatory approaches. Despite adopting a deductive approach, the creation

of categories was also influenced in part by the data (inductive). The terms associated with each model were notably informed by the work of Metcalfe (2019a,b; Table 1). The codebook, with the complete list of terms associated with each model, is available in "Material A" of the Supplement.

**Table 1. Simplified codebook with key terms used to code sampling units into corresponding model categories. Terms were directly used and adapted from Metcalfe (2019a,b). For coding, key terms shown below were used as a guide, exact term matches were not**
**required.**

| Focus | Deficit Model | Dialogue Model | Participatory Model |
|-------|---------------|----------------|---------------------|



| | | | |
|---|---|---|---|
| Objectives | - raise awareness <br> - educate <br> - inspire/excite <br> - promote geoscience careers | - help people make decisions <br> - make connections between people <br> - discover public opinion <br> - debate/discuss issues | - solve problems <br> - co-produce new knowledge <br> - participate in research with geoscientists <br> - participate in democratic policymaking |
| Activities | - one-way communication <br> - put up a display/exhibit <br> - use formal education to engage <br> - online means to communicate <br> - use traditional mass media | -activity involving people in geoscience <br> - train/develop skills to participate in geoscience <br> - workshops <br> -provide access to geoscientists | - collect data/do research <br> - jointly produce new knowledge <br> - participate with geoscientists in an activity |

Regarding the coding of activities, three supplementary categories (medium, resource, audience) were devised to extract more detailed insights from the data. This finer granularity enables a more precise depiction of the prevalence of models in practice and facilitates interpretations of models utilized in specific contexts, with particular resources, and for distinct audiences.

The medium category drew heavily from established venues described in the informal education context and created logical divisions in the activity data (Figure 1). The National Research Council (2009) identified four main settings for informal learning, namely everyday experiences, designed spaces, programs for science learning, and science media. While this content analysis primarily focused on venues other than "everyday experiences", these three remaining venues provided a structured framework for organizing and discussing the identified geoscience communication activities. Science media encompasses a

wide array of traditional and digital media formats distributed across all science learning venues. Programs for science learning typically occur within educational institutions and community-based organizations that prioritize science education. Designed settings, on the other hand, are intentionally crafted environments curated to facilitate learning and foster self-engaging experiences.

Mediums (science media, programs for science learning, designed settings) were each further categorized. Science media

includes traditional print media, traditional broadcast media, and new media. These categories were partially influenced by existing literature and natural breaks in the data (e.g. Rajendran & Thesinghraja, 2014). Within programs for science learning, workshops/training, supplemental resources, and festivals/science events inductively emerged as categorically distinct resources. Workshops/training are defined as "an in-person meeting where an individual/group explores a subject, develops a skill, or carries out a project". At first, the supplemental resources category was considered an "other" category, but during

preliminary coding, it was noted that it encompassed activities which reinforce, enrich, or extend understandings. Often it was




noted that these were offered in addition to a workshop/training. Festivals/events were typically characterized by one-off or special events that were unique from those activities associated with workshops/training.

Within designed settings, such as museums, Ahmad et al. (2015) further classified exhibits based on three criteria: the mode of apprehension, the type of learner, and their level of participation. They identified six distinct "types" of exhibitions:

aesthetic, didactic, hands-on, multimedia, minds-on, and immersive. Aesthetic & didactic exhibits include those where visitors apprehend through contemplation, reflection, text-based, cases, and murals. Hands-on exhibits involve those using low technologies and interactive activities, while minds-on & immersive exhibits encourage visitors to problem-solve and discuss and immerse themselves within the exhibition. Finally, audiences comprise K-12 students, teachers, and the general public. To aid in understanding the complex interrelationships between multiple categories and sub-categories, a concept map

illustrating these relationships has been provided (Figure 1).

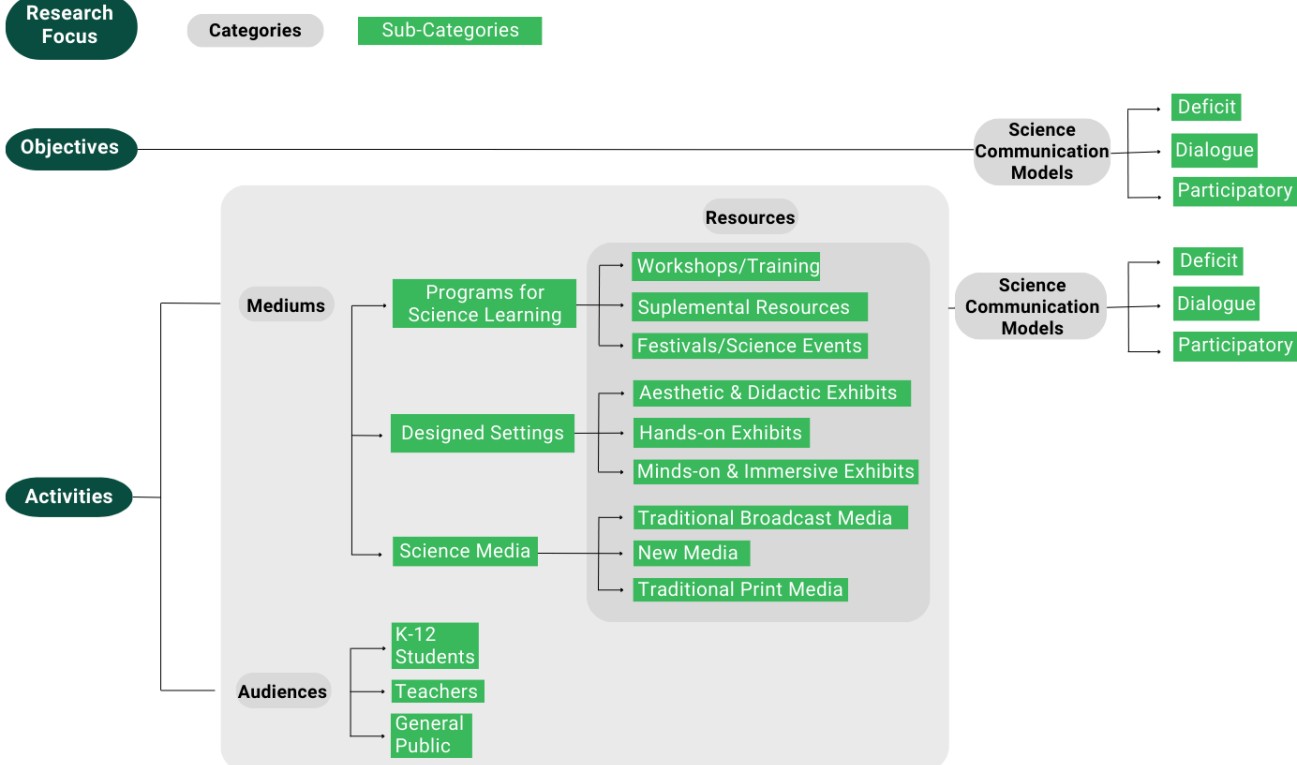

**Figure 1. Concept map highlighting the relationships between the categories, sub-categories, and research focuses.**





### 3.2.1 Recording/Coding

The data recording process, completed by the principal investigator, entailed multiple iterations aimed at defining the semantics of the data. This involved documenting words, phrases, images, and contextual observations pertaining to the activities and objectives of practitioners. The categories delineated in Figure 1 emerged as the outcome of these observations, evolving until they became exhaustive and mutually exclusive. In other words, each recording unit could be accommodated within a single category (Krippendorff, 2018). At times, it necessitated the creation of categories to encompass recording units that did not fit into existing ones. As detailed in Krippendorff (2018), each datum was systematically approached through a predefined sequence of decisions. This approach is often facilitated by codebooks, which serve as written protocols instructing coders on how to assign values to the content of interest (Lacy et al., 2015). In this study, the process can be divided into two parts.

The first part focuses on the practitioners' objectives, which proved relatively straightforward. Coders were instructed to locate the "Home" or "About" page on a practitioner's website and identify keywords associated with each model. For each occurrence of an objective per practitioner (to a maximum of three per model) coders checked off a box under the associated model in the coding database. The second part of the process centered on practitioners' activities. Coders examined all other pages on the website, starting from the home page and navigating through each tab. They noted keywords associated with each model and determined the target audience and resource for each activity. Coders were provided with a list of keywords associated with each model, derived from Metcalfe (2019a), with any new objectives or activities added to the codebook as necessary.

Once coders identified the associated audience and resource for each recording unit, they entered the corresponding code in the coding database. Coders were instructed to code only once for every unique combination of a resource-audience-model activity identified on a website. Additionally, to validate the coding process, coders provided qualitative context or evidence for the associated code by copying and pasting relevant data, such as words, phrases, or photographs. This step aided in ensuring consistency and reliability in the coding process, particularly when populating the full sample discussed in the Sect. 3.2.2 Reliability. Coders were encouraged to explain their thought process, if necessary, by enclosing their comments in asterisks.

### 3.2.2 Reliability

Given the subjective nature of applying codes to qualitative data, achieving intercoder reliability was crucial for this study. Reliability, denoting the trustworthiness of the data, was ensured through intra-coder agreement (stability) and inter-coder reliability (replicability). Intra-coder agreement was facilitated by several measures. First, the principal investigator, who also served as coder 1, ensured that coders were familiar with the terminologies and literature surrounding the science



communication models. Second, notes were maintained throughout the coding process to uphold consistency in coding instructions. Third, breaks were taken between coding each sample unit to prevent carelessness. Additionally, the data was coded multiple times with adjustments noted before commencing inter-coding.

Before conducting the intercoder reliability check, a reviewer was engaged to assess the readability of the coding instructions and the categories and sub-categories used in the database. This step proved invaluable as the reviewer identified important aspects overlooked by the lead researcher. For instance, the reviewer highlighted the importance of ensuring that intercoders had English as their first language to ensure familiarity with English idiomatic expressions. Moreover, the reviewer proposed an alternative approach to the coding process, leading to the development of version 2 of the coding instructions, which

streamlined the process significantly. While independent coders are typically involved in intercoder reliability checks, this study followed Lorr & McNair's (1966) cautionary note regarding the potential for coder bias. Thus, the reviewer was not engaged in the reliability check to avoid influencing the results.

In retrospect, it is acknowledged that knowledge of science communication models should have been a prerequisite for this study, as discussed later in this section. An advertisement was circulated to the undergraduate Earth science program at Simon

Fraser University, listing attention to detail, ability to follow rules, familiarity with Excel, and English as a first language as requirements. Basic knowledge of science communication and informal education were considered assets. Despite receiving four applications, only a single independent coder (coder 2) was hired due to budgetary and logistical constraints.

### The Test Sample

Before conducting the intercoder reliability tests, coder 2 underwent a training session to ensure consistency in coding

practices. Initially, background information on the study's content was reviewed, with careful attention to providing information solely from the literature rather than relying on the understanding of the codebook developer (coder 1). This step aimed to minimize the influence of coder 1's comprehension of the study content on coder 2. While knowledge of the study material is commonly recommended for intercoder reliability checks, it was not a qualification for coders in this instance. The rationale behind this decision was to enhance the clarity of coding instructions, thereby increasing the replicability of the study.

Unfortunately, the amount of time needed to train coder 2 to have a complete understanding of the science communication models was not feasible for this study. These time and budgetary constraints also meant that coder 2 was allotted only five hours for training on background information. Consequently, disagreements discussed in the subsequent section stemmed from misunderstandings of science communication models.

Following the background information training, coder 2 dedicated 12 hours to coding eight websites to populate the test sample.

Coder 1 supervised coder 2's coding of the first three sample units and asked coder 2 to articulate their thought process based on the codebook instructions while coding. This step enabled coder 1 to identify areas for improvement in the codebook, leading to the development of six versions of the codebook based on feedback from coder 2 and observations made by coder



1. The final version of the codebook (version 6; Material A of the Supplement) was used for developing the reliability sample and the full sample.

An intercoder reliability check was conducted on the test sample, and the complete table with results from these calculations at each categorical level for objectives and activities is available upon request. The agreement between coders 1 and 2 for the 63 objectives identified was 84% simple agreement, a Cohen's kappa of 0.64, and a Gwet's AC1 of 0.71. For the 567 activities identified, agreement was 87% simple agreement, a Cohen's kappa of 0.28, and a Gwet's AC1 of 0.85. While simple agreement was relatively high, Cohen's kappa and Gwet's AC1 provided additional insights into the reliability of the data. The primary

limitation of simple agreement lies in its failure to account for the potential random selection of codes (Carletta, 1996). While Cohen's kappa addresses this concern by incorporating chance into its computation, some scholars have raised reservations about its application, particularly when dealing with data featuring extreme marginal distributions (Dettori, 2020; Wongpakaran et al., 2013). Conversely, critiques of Gwet's AC1 highlight its leniency, fundamental methodological challenges, and the absence of a standardized classification system for its values (Vach & Gerke, 2023). This ongoing discourse

underscores the need for context-specific guidelines to aid in interpreting statistical agreements, as emphasized by Geiß (2021). In our analysis, we relied on Cohen's kappa statistics and its established thresholds (> 0.80 or > 0.70 for exploratory studies) to inform our reliability assessments (Intercoder Reliability, 2010; Landis & Koch, 1977).

Given the low Cohen's kappa values observed in the test sample and the high number of potential categories for a code to be applied to, combined with coder 2's lack of prior knowledge of science communication models, achieving "excellent"

intercoder reliability was deemed unfeasible. An alternative approach known as "double coding" was considered, where both coders code all data in the sample twice, as opposed to coding a subset of the total sample (Bogen et al., 2021; Fleerackers et al., 2022; Krippendorff, 2004, p. 250). This method was adopted to populate the reliability sample, considering its applicability to categorically complex cases (Spooren & Degand, 2010).

The Reliability Sample

The reliability sample used for intercoder reliability calculations comprised 729 objectives and 6561 activities from a total sample size of 81 websites. These reliability samples were generated as a result of the procedure outlined above, which involved initially encountering low intercoder reliability in the test sample (using Cohen's kappa as a reference) and subsequently conducting double coding for all samples. Table **2** presents the simple agreement, Gwet's AC1, Prevalence, Cohen's kappa, and Krippendorff's alpha for the three science communication model objectives. In this specific case,

interpretations of reliability should primarily rely on Gwet's AC1. The data exhibits the "Kappa paradox", where imbalanced marginal distributions and issues with agreement prevalence lead to low Kappa values (Delgado & Tibau, 2019; Tan et al., 2024; Wongpakaran et al., 2013; Zec et al., 2017). Since Krippendorff's alpha and Cohen's kappa are highly sensitive to this issue, they should not be used to interpret the reliability of the presented results. Given that Gwet's AC1 adjusts for chance





agreement and avoids the paradoxical behaviour of kappa, it is better suited for interpreting reliability. However, there are

currently no proposed benchmarks for interpreting the level of reliability of Gwet's AC1 and using the benchmarks proposed

by Landis & Koch (1977) for Gwet's values is not appropriate.

Within science communication objectives, the deficit model exhibited 72% observed agreement and a Gwet's AC1 value of

0.46 (Table **2**). The dialogue model showed 86% observed agreement and a Gwet's AC1 of 0.83, while the participatory model

demonstrated 88% observed agreement and a Gwet's AC1 of 0.85. In science communication activities, there were 81 unique

categories, and the intercoder reliability statistics can be found in Appendix A. Gwet's AC1 values for all categories ranged

from 0.56 to 1.00 (Mean= 0.92, SD= 0.10).

Despite the agreement results presented here, it's worth noting that the full sample discussed in the subsequent section

underwent a process of double coding, resulting in 100% agreement through a tie-breaker approach. Consequently, agreement

results from the full sample are typically not presented in discussions surrounding reliability, as all disagreements are resolved.

However, the relatively high values of Gwet's AC1 achieved at most categorical levels in the reliability sample indicate a

relatively high level of agreement between coders.

Table 2. Intercoder reliability statistics of science communication objectives in the reliability sample.

| Model Objectives | Ratings by Coders 1 and 2 | | | Prevalence | Observed Agreement | Gwet's AC1 | Cohen's Kappa | Krippendorff's Alpha |
|---|---|---|---|---|---|---|---|---|
| **Deficit Model** | | Present | Absent | Marginals | 25% | 72% | 0.46 | 0.42 | 0.41 |
| | Present | 61 | 18 | 79 | | | | | |
| | Absent | 51 | 113 | 164 | | | | | |
| | Marginals | 112 | 131 | 243 | | | | | |
| **Dialogue Model** | | Present | Absent | Marginals | 2% | 86% | 0.83 | 0.19 | 0.19 |
| | Present | 6 | 23 | 29 | | | | | |
| | Absent | 11 | 203 | 214 | | | | | |
| | Marginals | 17 | 226 | 243 | | | | | |
| **Participatory Model** | | Present | Absent | Marginals | 5% | 88% | 0.85 | 0.38 | 0.38 |
| | Present | 12 | 18 | 30 | | | | | |
| | Absent | 12 | 201 | 213 | | | | | |
| | Marginals | 24 | 219 | 243 | | | | | |

The Full Sample

The full sample represents the definitive dataset utilized in the subsequent results section and serves as the basis for all

interpretations. Transitioning from the reliability sample to the full sample involved enlisting an external expert to adjudicate

any discrepancies between coders 1 and 2. In this instance, the external expert was the supervisor of coder 1 (the lead





researcher). It's important to acknowledge this as a limitation, as the supervisor's perspectives may be influenced by their familiarity with the content through interactions with the lead researcher. Unfortunately, due to budget constraints, an alternative expert could not be engaged. To mitigate potential bias, the expert was kept blind to which coder had assigned a

specific code and could only rely on the qualitative context provided for each code to inform their decision-making process. In total, the expert resolved 558 disagreements between coders 1 and 2, employing a method commonly known as resolution by a tie-breaker (Lombard et al., 2002).

Among the 729 decisions made for objectives, there were 134 disagreements, while for activities, out of 6561 decisions, there were 432 disagreements. Relative to coder 2, the external expert agreed with coder 1 86.2% of the time for the deficit model

code, 72.7% for the dialogue model code, and 62.5% for the participatory model code in objectives. Overall, of the 134 disagreements, the external agreed with coder 1 81.8% of the time. Similarly, in activities, the external agreed with coder 1 95.6% of the time for the deficit model code, 97% for the dialogue model code, and 90.9% for the participatory model code relative to coder 2. Overall, of the 432 disagreements, the external agreed with coder 1 95.9% of the time. These figures indicate a pronounced tendency for the external expert to align closely with coder 1 and exhibit fewer disagreements with

them. The reported disagreements on model codes for activities may not necessarily reflect a disagreement on the specific model itself, but rather a disagreement on the audience or resource.

The full sample encompasses all agreements between coders 1 and 2 (596 for objectives, 6137 for activities), agreements between the external expert and coder 1 (63 for objectives, 283 for activities), and agreements between the external expert and coder 2 (14 for objectives, 12 for activities). Disagreements between the external expert and either coder 1 or coder 2 (12 for

objectives with coder 1, 30 for activities with coder 1, 45 for objectives with coder 2, and 107 for activities with coder 2) were excluded from the full sample, as they indicated discrepancies between at least one coder and the external expert.

## 4 Results

Out of the 155 geoscience communication objectives identified in the coding process, an overwhelming 76% were deficit, while objectives aligning with the dialogue and participatory models constituted 11% and 13% of the data, respectively (Figure

**2**). Within the deficit model, objectives relating to education were predominant. Conversely, among those coded as dialogue, objectives focused on fostering connections between individuals. Lastly, within the participatory model, objectives centred around problem-solving emerged as the most common. In terms of geoscience communication activities, out of the 363 identified, a majority (61.4%) were coded as deficit, with dialogue activities comprising 33.3%, and participatory activities making up 5.3% of the total (Figure **2**).

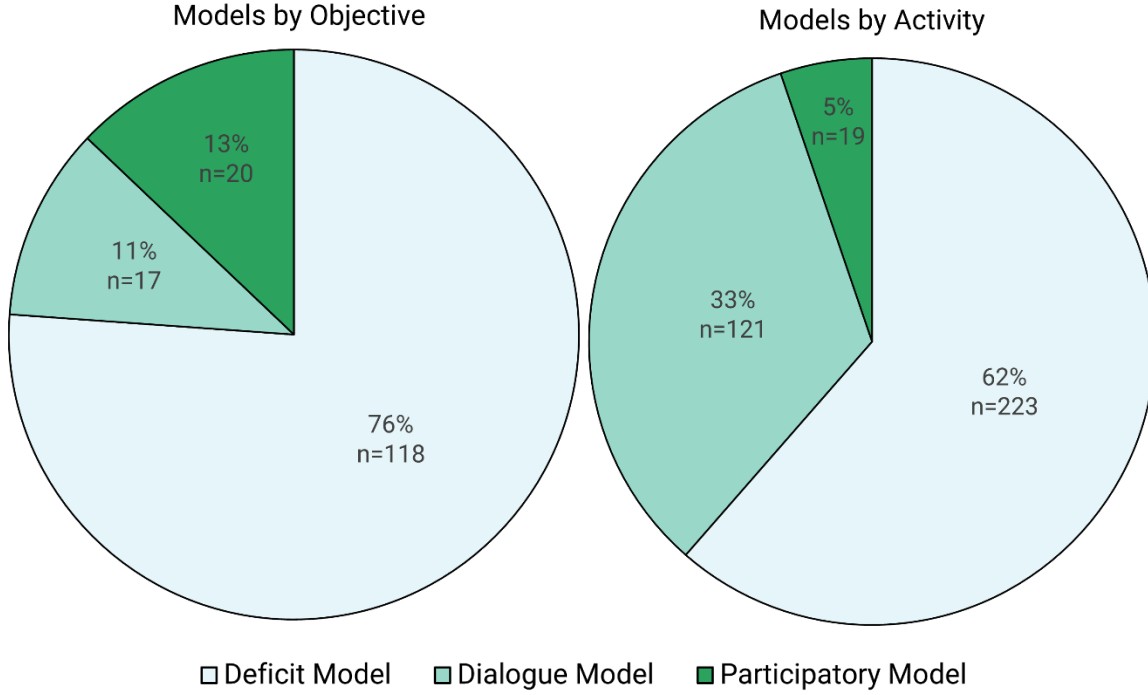

Figure 2. Pie charts visualizing relative proportions of deficit, dialogue, and participatory model codes applied to objectives and activities.

The deficit model was used most for activities targeting the general public, representing 68% of all activities, while it was least utilized for K-12 students, accounting for 48% of activities. Dialogue model activities were predominantly employed for K-12 students and utilized least for the general public. Notably, the participatory model only comprised 2-6% of all activities and was most commonly used for K-12 students and general public audiences. In Figure **3**b, it is evident that
science media predominantly featured deficit model activities, encompassing 83% of all activities coded. Programs for science learning exhibited the highest proportions of the dialogue and participatory models, while designed settings were relatively evenly distributed between the deficit and dialogue models. Among the mediums, designed settings had the fewest activities coded in total, with 50% coded as deficit, 39% as dialogue, and 1% as participatory.



**Models by Audience**

**100% Models by Audience**

 a)

**Models by Medium**

**100% Models by Medium**

b)



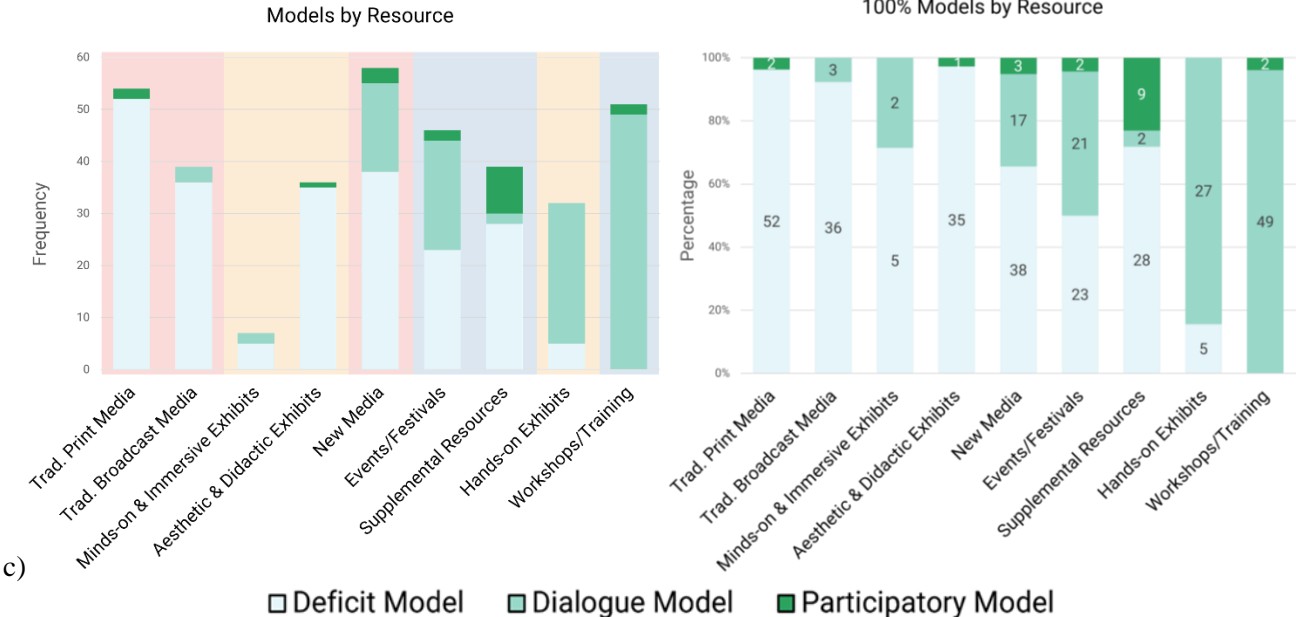

c)

Figure 3 Respective frequencies and distributions of model activities used a) when communicating with target audiences, b) in science communication mediums, and c) resources (sorted by increasing participation from left to right). The colours in c correspond to the medium (red/orange = science media, yellow = designed settings, blue = programs for science learning).

## 4.1 Model Activities in Resources:

This section has been structured to emphasize the distribution of deficit, dialogue, and participatory models across resources
and mediums: science media (Table **3**), programs for science learning (Table **4**), and designed settings (Table **5**). Qualitative data (e.g. words, images) associated with the theoretical models, gathered during the coding process, provided deeper insights into their practical implementations. Each model and resource is accompanied by examples of excerpts used for coding specific models. The absence of a model under a resource indicates that no activities were coded to that particular model.

### 4.1.1 Science Media

In the majority of cases, traditional print media activities were coded as the deficit model, accounting for 96% of observed codes (Figure **3**c). Two activities employing a co-creation approach were also coded as participatory. These activities encompassed lesson plans, books, newsletters, and other written media forms (Table 1). The deficit and dialogue models respectively accounted for 92% and 8% of activities coded to traditional broadcast media (Figure **3**c) with YouTube videos, movies, and slideshows being the most common (Table **3**). If any of these resources were used for training purposes, they were
additionally coded as dialogue. Activities coded to new media were primarily deficit (66%), followed by dialogue (29%) and participatory (5%) models (Figure **3**c). Deficit activities included podcasts, blogs, apps, and websites/platforms. If activities



were coded to workshops/training within programs for science learning or hands-on exhibits within designed settings, and were offered virtually, they would additionally be coded as dialogue. Lastly, any apps or data/web platforms that were used for citizen science were coded as participatory (Table **3**). Concerning audiences, traditional print media activities were

distributed relatively evenly among the general public, K-12 students, and educators, while traditional broadcast media and new media activities were most frequently coded for the general public (Figure **4**a).

Table 3. Common activities coded and examples of data used to code models within resources of science media. Bolded text in the "Example" column corresponds to words/phrases associated with key terms for coding models, and relevant anchor samples for determining the corresponding resource. "*" under the example column is derived from coder observations.

| Resource | Model | Activities | Example |
|---|---|---|---|
| Traditional Print Media | Deficit | - lesson plans<br>- books<br>- newsletters | "The new Mining Matters **Activity Book** for youth…" – Mining Matters |
| | Participatory | - co-created print media | "Ocean Sense core modules, **co-created** by ONC and Indigenous community partners… download **lesson plans** and activities, and browse connections to curriculum". – Ocean Networks Canada |
| Traditional Broadcast Media | Deficit | - YouTube videos<br>- Slideshow<br>- movies | "Canadian Mining **Videos**" *various videos relating to mining in Canadian society – Mining Association of Canada |
| | Dialogue | - training videos | "Our Deeper and Deeper **video tutorials** are now available!" *within a tab related to resources for teachers – Mining Matters |
| New Media | Deficit | - podcasts<br>- blogs<br>- apps<br>- websites/platforms | "epicenters of local earthquakes as they are detected and located, are illustrated on a simple **web interface**" - SchoolShakes |
| | Dialogue | - virtual workshops<br>- virtual field trips/tours<br>- virtual training | "We are offering both in-person and **virtual outreach** for the 2023-2024 school year. All **workshops** are approximately…" -Let's Talk Science (UBC Okanagan) |
| | Participatory | - citizen science apps/platforms | "Welcome to our **data platform**! **Collect and share water quality data**". – Water Rangers |

**4.1.2 Programs for Science Learning**

Primarily, workshops/training inherently involve a certain level of interactivity, explaining the higher prevalence of activities coded as dialogue (96%; Figure **3**c). Workshops, field trips, courses, and hands-on activities were the most common examples of the dialogue model in practice, catering to the general public, teachers, and K-12 students (Table **4**, Figure **4**b). Additionally, two activities classified under the participatory model were identified, including a participatory professional development

workshop and a community training initiative, both offered by the same practitioner (Table **4**). Within supplemental resources, activities were coded to all models with the deficit and participatory models accounting for 72% and 23% respectively (Figure



**3**c). Deficit model activities included test/sample kits and games, while participatory activities solely included citizen science initiatives (Table **4**). Furthermore, two dialogue model activities were coded, including a poll researching public opinion on mining and a virtual research challenge on climate change (Table **4**). Events/festivals were evenly distributed between the

deficit and dialogue models (50%, 46%; Figure **4**b) constituting guest speakers for the former and camps and interactive community events for the latter (Table **4**). Two participatory activities were identified including a private paleontological dig and a camp centred around integrating Indigenous and Western science perspectives on water (Table **4**). Notably, only one event targeting teachers was identified, which was a professional development opportunity held at a mining conference (Figure **4**b).

Table 4. Activities and examples of data used to code models within resources of programs for science learning.

| Resource | Model | Activities | Example |
|---|---|---|---|
| Workshops/ Training | Dialogue | - workshops<br>- training/courses<br>- field trips<br>- hands-on activities | "An important component of each **workshop** is the package of resource materials provided to each participating teacher for use in the classroom" - EdGeo |
| | Participatory | - participatory teacher training<br>- participatory training | "educators may also participate in at sea expeditions where they **work alongside scientists**, engineers, and technicians" – Ocean Networks Canada |
| Supplemental Resources | Deficit | - test/sample kits<br>- games<br>- database | "this **kit** provides an introduction to the basics of geology. Supplied in the kit are over 40 mineral specimens, testing kits, and examples of prospecting equipment". – Rossland Museum & Discovery Centre |
| | Dialogue | - public polls<br>- research challenges | "releasing a new **national poll** that finds high levels of support for Canadian mining and increased understanding on the role Canada's mining industry…" – Mining Assoc. of Canada |
| | Participatory | - citizen science | "**Citizen Science** Initiatives…The CNHR is pleased to work with interested members of the public to **answer research questions and develop tools** to enhance our understanding and ability to respond to natural hazards". – SFU CNHR |
| Events/ Festivals | Deficit | - guest speakers | "Our outreach program **provides presenters to your classroom** to teach your geoscience curriculum" – Burgess Shale Foundation |
| | Dialogue | - camps<br>- special one-off activities<br>- interactive community events | "In Dinosaur Day **Camps,** kids (ages 7 – 13) learn the same skills used by actual paleontologists to find, clean, and learn about fossils!" – Tumbler Ridge GeoPark |
| | Participatory | - participatory events | "Our **private digs** are global adventures, so make sure you have your passport and a hunger for new experiences!" *Image of people **participating with scientists** during fossil dig - DinoLab |



### 4.1.3 Designed Settings

The phrase "put up a display/exhibit" was associated with the deficit model according to Metcalfe (2019a,b) leading to the majority of activities in designed settings being coded as deficit. Nevertheless, due to the interactive nature of many tours and workshops offered in designed settings, activities aligned with the dialogue model were also identified. Aesthetic/didactic exhibits were overwhelmingly coded as deficit activities (97%; Figure **3**c) with collections, displays, and interpretive signage as the most common activities with most intended for general public audiences (Figure **4**; Table **5**). A collection that was co-produced with members of the public was coded as participatory (Table **5**). Hands-on exhibits were largely coded as dialogue (84%, Figure **3**c) since tours and workshops where participants were involved in science were common (Table **5**). The deficit model accounted for 16% of coded activities and included interactive displays utilizing low technology to communicate (Figure **3**c; Table **5**), typically targeting K-12 students and the general public (Figure **4**). Lastly, minds-on & immersive exhibits were less frequently coded overall (Figure **3**c). Among those identified, 71% were coded as deficit including immersive displays and 29% were coded as dialogue which included immersive tours (Figure **3**c; Table **5**).

Table 5. Activities and examples of data used to code models within the resources of designed settings.

| Resource | Model | Activities | Example |
|---|---|---|---|
| Aesthetic/ Didactic Exhibits | Deficit | - collections<br>- displays<br>- interpretive signage | "six interactive posts with **educational panels** featuring the history of mining in the Elk Valley with imagery reflecting the geological roots of mining" – Tourism Fernie |
| | Participatory | - co-produced collections | "New specimens (FOSSILS) are added to the collection through museum-led field expeditions, **donated discoveries by residents** from across British Columbia…" – Royal B.C. Museum |
| Hands-on Exhibits | Deficit | - interactive displays | "This spherical **interactive display** projects images and animations of planets, real time weather, ocean currents" – Pacific Museum of Earth |
| | Dialogue | - tours<br>- workshops | "Dinosaur Trackway **Tours**: Experience 97 million year old dinosaur footprints up close in their natural environment!" – Tumbler Ridge Museum |
| Minds-on & Immersive exhibits | Deficit | - immersive displays | "**Gaia Gallery**…A trail of giant footprints leads you to an enormous, armoured goliath that once walked these lands. **Immerse** yourself in this ancient world and imagine what it was like to live during this time". – the Exploration Place |
| | Dialogue | - immersive tours | "**Exhibits**…BOOM! -- a **live-action experience** inside the historic Mill…will take visitors on thrilling visual journey exploring all 20-storeys…" – Brittania Mine Museum |



Figure 4 Frequency of deficit, dialogue, and participatory model activities identified in a) Science Media, b) Programs for Science Learning, and c) Designed Settings relative to target audiences.



## 4.2 Model Activities by Region


Based on the primary region of services offered, the distributions of activities offered across B.C. were as follows: Lower Mainland/Sea-to-Sky 28%, online only (available to all regions) 19%, Canada-wide (available to all regions) 12%, Vancouver Island 16%, Thompson-Okanagan 6%, Kootenays 8%, Northern B.C. 8%, Canyons and the Cariboo 2%, and North & Central Coast 1%. Across all geographic locations (excluding online and nation-wide activities), the range in proportions of activities

coded to the science communication models were: deficit (46-66%), dialogue (31-55%), and participatory (0-5%). Figure **5** provides the exact proportions of coded model activities for each region. While quantitative outcomes are presented, it is important to exercise caution in interpreting these numbers, given that activities were solely categorized based on target audience, medium, and resource type. Therefore, precise figures and percentages should only be cited for these specific analyses. Nonetheless, general trends can still be explored, as the regions where practitioners offered their services were

documented during the database construction. Here, it is assumed that all activities offered by a practitioner are delivered within their primary region of service provision.






Figure 5. Map visualizing regions of British Columbia and corresponding deficit, dialogue, and participatory activities offered. Pie chart sizes are approximately proportional to the region's population. The n value corresponds to the number of recording units identified in respective regions.



## 4.3 Models by Venue

The venues identified were categorized using a combination of classifications found in the literature, as discussed in Sect. 3 Methods, and categories that emerged from the data itself. The distributions of activities based on their primary venue are as follows: 23% are available online, 21% in history museums, 17% in science museums, 14% in natural physical sites, 14% in

K-12 schools, 7% in parks, and 4% in universities. The deficit model's use ranged from 29-71%, the dialogue model from 22-71%, and the participatory model from 0-23% across all venues. Deficit objectives and activities were most commonly observed in organizations associated with universities and parks, as illustrated in (Figure **6**). Conversely, dialogue objectives and activities were predominantly found in online platforms and university environments. Finally, participatory objectives and activities were most frequently encountered in organizations operating within natural physical sites for both examined cases.

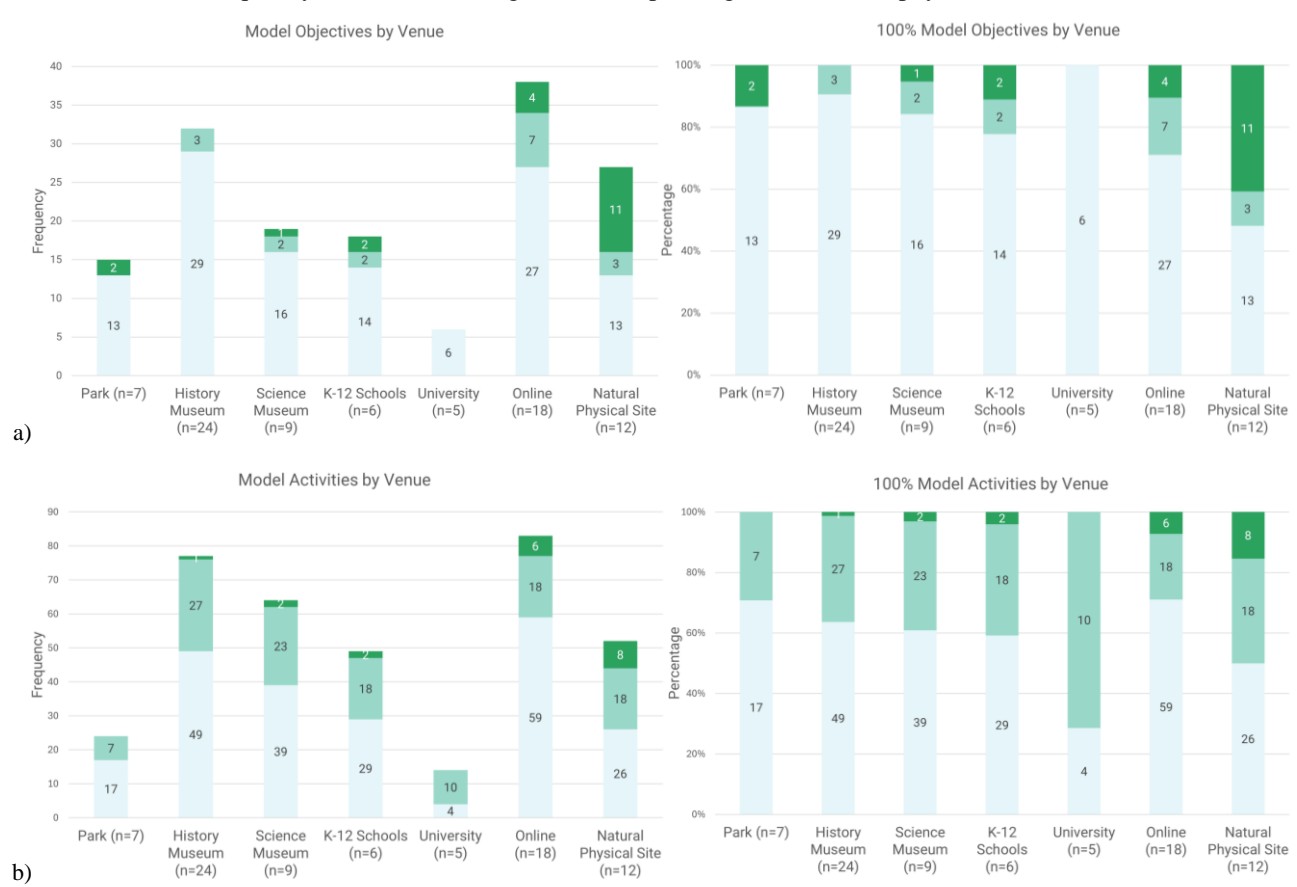

Figure 6. Respective distribution of model objectives (a) and activities (b) when communicating in select venues.





## 5 Discussion

The deficit model accounts for 61% of activities and 76% of objectives coded, suggesting its persistent influence in geoscience communication practices. Despite the identification of some promising participatory activities and objectives, constituting only 5% (n=19) and 13% (n=20) of the overall practice, respectively, it's evident that the deficit model still predominates. These findings may offer a broad understanding of geoscience communication practices in Canada, but they are primarily representative of British Columbia. This study does not intend to represent global geoscience communication practices or broader science communication landscapes, which are shaped by a myriad of socio-economic, geopolitical, historical, and cultural factors (Bauer et al., 2007; Gascoigne et al., 2020; Horst, 2012).

The data shows a misalignment between practitioners' stated objectives and the activities they design to achieve them. Deficit and participatory objectives do not consistently translate into deficit and participatory activities. Alternatively, dialogue activities are used to address these objectives. This may be attributable to several factors (e.g. the target audience, communicators' predispositions). For instance, when K-12 students are the audience, *hands-on activities* (dialogue activity) are often used *to educate* (deficit objective) resulting in a misalignment. More deficit objectives were coded compared to activities suggesting that practitioners believe deficit objectives are more important than participatory objectives. Regarding the overrepresentation of participatory objectives relative to activities, this could be due to 1) the comprehensive nature of many participatory activities, allowing a single activity to meet multiple participatory objectives, or 2) a belief that deficit and dialogue activities can still achieve these objectives.

Beyond the factors mentioned above, it is also possible that practitioners are unaware of the full range of objectives they could aim to address, or have not considered the alignment of their objectives and activities. While we highlight these misalignments, we do not necessarily view them as problematic, especially when more participatory activities are used for less participatory objectives. However, practitioners may be selling themselves short in this case. Conversely, using less participatory activities to meet more participatory objectives may present more significant issues. For example, it is unlikely that *lecturing* (deficit activity) would allow practitioners *to gain the public's involvement in democratic policymaking* (participatory objective).

The deficit model is used most when communicating with general public audiences, and least with K-12 students. This finding along with the prevalence of the dialogue model when communicating with K-12 students aligns with the notion that hands-on activities are particularly effective in engaging youth in an educational context. Notably, participatory activities, such as citizen science, are more prevalent among general public audiences, while educators appear to have limited access to such activities. Given that educators regularly engage with diverse student cohorts and considering the participatory model's potential for meaningful audience engagement, prioritizing participatory activities for educators could offer a promising avenue forward. However, it is essential to acknowledge that educators have specific learning outcomes required from the curriculum, and other constraints (e.g. time, funding, resources) which should inform the development of suitable opportunities





for them. These constraints may partially account for the prevalence of deficit and dialogue activities, where time considerations are less critical.

Programs for science learning exhibit the highest proportions of dialogue and participatory activities, constituting 63% of all
coded activities. These programs predominantly encompass citizen science initiatives and hands-on workshops, respectively. Beyond citizen science endeavours, the participatory model demonstrates its capacity to engage diverse audiences within designed settings, as evidenced by "fourth-generation science museums" featuring co-created and equitable exhibits (De Oliveira, 2023; Pedretti, 2020). Among designed settings, aesthetic and didactic exhibits are most prevalent in British Columbia, likely influenced by the province's rich mining history, which often finds commemoration through museum exhibits
and interpretive signage. Hands-on exhibits primarily fall under the dialogue model, especially when accompanied by interactive workshops and guided tours. Conversely, science media activities are predominantly coded as deficit, reflecting the inherent deficit orientation of traditional media formats. However, an amendment to the codebook facilitated the identification of additional dialogue and participatory activities within science media underscoring the importance of contextual information in coding decisions. For example, the emergence of virtual workshops for K-12 students in new media formats, likely in
response to the Covid-19 pandemic, has contributed to the increased prevalence of the dialogue model in this medium. New media formats offer more opportunities for interactive engagement compared to traditional media, which may explain the higher frequency of activities coded here. Nevertheless, creating science media activities may be perceived as easier, less time-consuming, and more cost-effective than developing activities within other mediums. We've also shown that science media activities can occur across all venues unlike designed setting activities. These factors likely explain the high frequency of
activities coded to science media relative to designed settings.

Regarding the primary geographic locations of services offered, it's noteworthy that targeted participatory activities are notably lacking. The majority of participatory activities originate from online-only or nationwide practitioners, rather than region-specific initiatives within British Columbia. Focused participatory activities tailored to specific regions are likely to have a significant impact compared to online-only and nationwide initiatives. Despite an overall underrepresentation of the
participatory model across all regions of British Columbia, the absence of targeted participatory activities in the Lower Mainland/Sea-to-Sky region, which boasts the highest population in the province, is particularly striking. Examination of barriers and the addition of incentives may promote future participatory initiatives. Moreover, considering regions where geoscience intersects with the public, such as communities close to mining, oil and gas, and other land or water use issues, could enhance opportunities for targeted geoscience communication.

Across various venues for geoscience communication, deficit model activities are most prevalent in parks and history museums, predominantly through aesthetic and didactic exhibits and traditional print media. Conversely, dialogue and participatory models are more common in natural physical sites and science museums, where hands-on activities and citizen science programs thrive. The participatory model's prevalence in natural physical sites is likely attributed to its capacity for



place-based engagement, particularly in activities involving citizen science data collection in water bodies. This observation underscores the importance of considering the contextual relevance and effectiveness of different communication models across diverse geoscience communication settings.

## 5.1 Why Does the Deficit Model Persist?

The persistence of deficit model objectives and activities may imply that practitioners and the organizations facilitating these opportunities are either unaware of participatory models or lack training in their utilization. Gascoigne et al. (2020) discuss the limited availability of science communication training in Canada, with only one master's program in Ontario and few other universities offering courses as part of accredited programs. A recent study by Vickery et al. (2023) shared a similar objective to this research. Instead of quantifying the use of science communication models in geoscience communication practice, they assessed the models' utilization in science communication training for post-secondary STEM students. Through a coding process of published research on science communication training, they found that 40.7% of the terminology reflected the deficit model, 39.5% the dialogue model, and 19.8% the participatory model. Comparing these results to those of our study, we observe a consistent trend of the participatory model being underrepresented. If the participatory model is marginalized in the training of science communicators, it follows that its representation in practice would be even further diminished. Research has demonstrated that scientists' attitudes toward communication often influence their communication practices (Besley et al., 2018; Kessler et al., 2022). For example, findings such as those reported by Calice et al. (2022), which indicate persistent deficit-oriented views of science communication, suggest that deficit activities are likely to prevail, thus providing further support for our findings. The prevalence of the deficit orientation is further accentuated by institutional structures and the model's utility for public policy purposes (Simis & Madden et al., 2016). Another consideration pertains to the challenges associated with implementing the participatory model in practice. Despite the numerous advantages associated with the participatory model, it is crucial to acknowledge its limitations. These initiatives are often characterized by high costs, time-intensive processes, a tendency to engage the scientifically literate, and limited reach, among other factors (Barbosa-Gómez et al., 2022; Nisbet & Scheufele, 2009; Powell and Colin, 2009). Of the participatory activities identified in this research, most were associated with practitioners communicating on atmospheric science, hydrology, and oceanography, which arguably have a more obvious impact on people and everyday life. This finding aligns with existing literature which suggests that the participatory model tends to suit problem science rather than basic science (Callon, 1999; Metcalfe, 2019b).

While a significant portion of the science communication literature portrays the deficit model in a negative light (Macq et al., 2020; Nisbet & Scheufele, 2009), others have highlighted its potential value in science communication practice (Stoker & Tusinski, 2006; Trench, 2008). For instance, Stoker & Tusinski (2006) propose that the deficit model can foster responsibility, diversity, and reconciliation. It's worth noting that in many instances, the deficit model effectively achieves its objectives.



Trench (2008), for example, highlights the success of Richard Dawkins through his bestselling book and other deficit-driven
       activities. The model's efficacy was also demonstrated in addressing the public's need for information, as exemplified during
       the COVID-19 pandemic (Zimmerman, 2024). On the other end of the spectrum, the participatory model is theorized to
       encourage inclusive involvement in science and democracy and build long-term relationships (Borchelt & Hudson, 2008;
       Schrögel & Kolleck, 2019). Recent studies, including Orthia et al. (2021), provide emerging practical evidence of the model's

benefits. These studies illustrate how co-design can enhance engagement and foster a sense of inclusion and shared identity.
       Moreover, community engagement can empower stakeholders to make informed decisions, while community partnerships can
       yield direct positive health outcomes.

## 6 Limitations

       Considerable limitations accompany this study and warrant acknowledgment. Firstly, the study was confined to publicly

available websites, excluding initiatives without online presence. This decision, made to limit the study's scope, disregards the
       impact of those practitioners and their offerings. Notably, social media influencers and geo-artists were excluded as a result.
       The keywords used in the data search and database population may reflect biases, possibly omitting terms used in other
       countries. This inherently adds the possibility of certain practitioners not being included in the database in the first place.
       The database primarily encompasses practitioners in British Columbia, with the additional inclusion of practitioners from

across Canada if they offered online resources or programs in British Columbia. While it's probable that some practitioners
       from Canada and British Columbia are not represented in the database, we believe that most organizations with a website
       presence (other than on social media are included), including both those with significant reach in terms of public-facing
       geoscience resources, and those with more local and targeted reaches have been included.
        Moreover, the study restricts itself to terms associated with science communication models based on Metcalfe (2019). This

narrow focus overlooks alternative terms linked to these models. This limitation also restricted the coding of particular
       resources and mediums. For example, "put up a display/exhibit" was to be coded as a deficit activity according to the codebook,
       meaning every activity in designed settings and its corresponding resources should only be coded as deficit. Even with the rule
       to override a particular model code (if context from another model is provided; which in itself is a limitation), it is possible
       that the quantitative results collected were skewed towards those corresponding models.

Furthermore, concerning the coding of activities to a single model, it is probable that combinations of these models are
       employed in practice (Brossard & Lewenstein, 2010; Jensen & Holliman, 2016; Metcalfe, 2019b). For instance, workshops
       for K-12 students typically begin with a lecture-based component, which is then followed by hands-on activities. However, in
       our analysis, the term "workshop" was coded solely as a dialogue activity, even though these workshops undoubtedly included
       deficit activities as well. This highlights a misalignment in the categorization of activities, where the current coding scheme



does         not         fully         capture         the         multifaceted         nature         of         science         communication.
In considering the shift from a deficit to a dialogue model, we implicitly assume that historical geoscience communication
practices were primarily characterized by deficit model approaches. Although this study lacks a temporal dimension, we
operate under the assumption that deficit model communication predominated in previous Canadian geoscience
communication practices (e.g., Schiele, 2008).

The transformation of a continuous variable to a binary variable results in a loss of some information. With this in mind, the
quantitative data presented does not accurately represent all activities available. For example, if a practitioner had four copies
of a book, this would only be coded once. It was often observed that this limitation occurred with deficit activities, potentially
resulting in an underrepresentation of the deficit model. Although intercoder reliability on proposed categories is typical of
numerous studies (Krippendorff, 20024), it was not conducted as part of the content analysis in this research. Significantly

more training would be necessary for the second coder to assess the suitability of applied categories, and with budgetary
constraints, this was not deemed feasible. Another limitation arose with the structure of the database when performing
intercoder reliability tests. For instance, a website was coded with three participatory objectives by one coder, but only two
participatory objectives by the other coder. When resolving discrepancies like this (via tie-breaker), the external can only be
guided by the qualitative data provided for the code of interest. Even if that code was applied already by the other coder under

one of the two other columns for participatory objectives, it could still get coded again (if the external agreed), thus
overrepresenting the coded model of interest. Lastly, the external expert, serving as the supervisor of coder 1 (the lead
researcher), may have been influenced in their perspectives on models due to their interactions with the lead researcher.

## 7 Conclusion

Findings from our content analysis of geoscience communication objectives and activities in British Columbia suggest that the

deficit model persists while the participatory model is significantly underrepresented in practice. Therefore, the shift from
"deficit to dialogue" commonly referenced in science communication literature has not been reflected in geoscience
communication practice, particularly in the context of British Columbia, Canada.

We theorize that the identified misalignments between practitioners' objectives and activities may result from adherence to
conventional objectives, a belief that activities and objectives do not need to be aligned, or simply a lack of consideration of

these factors. Regarding the target audience, it appears that the deficit model is predominantly used for communicating with
the general public, while the dialogue model is primarily employed for K-12 students. Few participatory model activities were
offered for educators, indicating a significant opportunity for future work.





While limitations with the terms used to code activities in designed settings may have overemphasized the deficit model, it is evident that these settings predominantly host deficit and dialogue activities. Programs for science learning exhibited the highest proportions of dialogue and participatory activities, whereas deficit activities dominated science media.

From an accessibility standpoint, participatory activities are greatly underrepresented across British Columbia. Many participatory offerings were part of nation-wide or online-only initiatives. We hypothesize that these would have less impact compared to targeted, community-specific, participatory programming, and therefore there is room for improvement in future practice. Lastly, concerning the venues where communication occurs, it was evident that parks (e.g. national, provincial, GeoParks) were dominated by deficit model communication, while natural physical sites (e.g. bodies of water, backyards) provided greater opportunities for dialogue and participatory model activities.

The findings discussed above provide a theoretical framework for further research and practice in geoscience communication. Additionally, they highlight areas for increased attention moving forward, such as training for practitioners, which can enhance geoscience communication offerings. There are numerous opportunities to expand on the research findings presented here. For example, evaluating and assessing the impact of geoscience communication practice could lead to more effective communication strategies. Furthermore, understanding how institutional/organizational factors, resource allocations, audience perceptions, virtual versus in-person programming, and cultural/contextual factors relate to the use of particular science communication models would be valuable avenues for future research.

**Appendices**

**Appendix A.** Table of intercoder reliability statistics between coders 1 and 2 for science communication activities for the reliability sample. Categories with a statistic of "undefined" or "1.00" indicate no data was coded to this category.

| Resource | Audience | Model Activities | Prevalence | Observed Agreement | Gwet's AC1 | Cohen's Kappa | Krippendorff's Alpha |
|---|---|---|---|---|---|---|---|
| Traditional Print Media | K-12 Students | Deficit | 0% | 0.83 | 0.79 | -0.04 | -0.09 |
| | | Dialogue | 0% | 1.00 | 1.00 | undefined | undefined |
| | | Participatory | 0% | 1.00 | 1.00 | undefined | undefined |
| | Teachers | Deficit | 0% | 0.77 | 0.70 | -0.05 | -0.13 |
| | | Dialogue | 0% | 0.99 | 0.99 | 0.00 | 0.00 |
| | | Participatory | 0% | 0.99 | 0.99 | 0.00 | 0.00 |
| | General Public | Deficit | 14% | 0.83 | 0.74 | 0.52 | 0.50 |
| | | Dialogue | 0% | 0.96 | 0.96 | 0.00 | -0.01 |
| | | Participatory | 0% | 0.96 | 0.96 | 0.00 | -0.01 |
| | | Deficit | 0% | 0.86 | 0.84 | 0.00 | -0.07 |





| | | | | | | | |
|---|---|---|---|---|---|---|---|
| Traditional Broadcast Media | K-12 Students | Dialogue | 0% | 1.00 | 1.00 | undefined | undefined |
| | | Participatory | 0% | 1.00 | 1.00 | undefined | undefined |
| | Teachers | Deficit | 1% | 0.96 | 0.96 | 0.38 | 0.38 |
| | | Dialogue | 0% | 0.96 | 0.96 | -0.02 | -0.01 |
| | | Participatory | 0% | 0.99 | 0.99 | 0.00 | 0.00 |
| | General Public | Deficit | 5% | 0.73 | 0.61 | 0.17 | 0.11 |
| | | Dialogue | 0% | 0.98 | 0.97 | -0.01 | -0.01 |
| | | Participatory | 0% | 1.00 | 1.00 | undefined | undefined |
| New Media | K-12 Students | Deficit | 0% | 0.86 | 0.84 | 0.00 | -0.07 |
| | | Dialogue | 0% | 0.90 | 0.89 | -0.02 | -0.05 |
| | | Participatory | 0% | 0.99 | 0.99 | 0.00 | 0.00 |
| | Teachers | Deficit | 0% | 0.93 | 0.92 | -0.03 | -0.03 |
| | | Dialogue | 0% | 0.96 | 0.96 | 0.00 | -0.01 |
| | | Participatory | 0% | 0.99 | 0.99 | 0.00 | 0.00 |
| | General Public | Deficit | 4% | 0.74 | 0.64 | 0.09 | 0.07 |
| | | Dialogue | 1% | 0.89 | 0.87 | 0.15 | 0.13 |
| | | Participatory | 1% | 0.94 | 0.93 | 0.26 | 0.26 |
| Workshops/ Training | K-12 Students | Deficit | 0% | 0.99 | 0.99 | 0.00 | 0.00 |
| | | Dialogue | 7% | 0.79 | 0.70 | 0.34 | 0.29 |
| | | Participatory | 0% | 0.88 | 0.86 | 0.00 | -0.06 |
| | Teachers | Deficit | 0% | 0.94 | 0.93 | 0.00 | -0.03 |
| | | Dialogue | 5% | 0.88 | 0.85 | 0.38 | 0.38 |
| | | Participatory | 0% | 0.94 | 0.93 | -0.02 | -0.03 |
| | General Public | Deficit | 0% | 0.95 | 0.95 | 0.00 | -0.02 |
| | | Dialogue | 4% | 0.77 | 0.68 | 0.16 | 0.11 |
| | | Participatory | 0% | 0.89 | 0.88 | 0.00 | -0.05 |
| Supplemental Resources | K-12 Students | Deficit | 0% | 0.83 | 0.79 | 0.00 | -0.09 |
| | | Dialogue | 0% | 0.95 | 0.95 | -0.02 | -0.02 |
| | | Participatory | 0% | 0.95 | 0.95 | 0.00 | -0.02 |
| | Teachers | Deficit | 1% | 0.95 | 0.95 | 0.32 | 0.31 |
| | | Dialogue | 0% | 0.99 | 0.99 | 0.00 | 0.00 |
| | | Participatory | 0% | 1.00 | 1.00 | undefined | undefined |
| | General Public | Deficit | 1% | 0.89 | 0.87 | 0.15 | 0.13 |
| | | Dialogue | 0% | 0.94 | 0.93 | -0.03 | -0.03 |
| | | Participatory | 5% | 0.93 | 0.91 | 0.53 | 0.53 |
| | | Deficit | 0% | 0.88 | 0.86 | -0.02 | -0.06 |





| | | | | | | | |
|---|---|---|---|---|---|---|---|
| Festivals/ Events | K-12 Students | Dialogue | 0% | 0.88 | 0.86 | -0.02 | -0.06 |
| | | Participatory | 1% | 1.00 | 1.00 | 1.00 | 1.00 |
| | Teachers | Deficit | 0% | 1.00 | 1.00 | undefined | undefined |
| | | Dialogue | 0% | 0.99 | 0.99 | 0.00 | 0.00 |
| | | Participatory | 0% | 1.00 | 1.00 | undefined | undefined |
| | General Public | Deficit | 0% | 0.83 | 0.79 | -0.06 | -0.09 |
| | | Dialogue | 0% | 0.90 | 0.89 | 0.00 | -0.05 |
| | | Participatory | 0% | 0.99 | 0.99 | 0.00 | 0.00 |
| Aesthetic & Didactic | K-12 Students | Deficit | 0% | 0.94 | 0.93 | -0.03 | -0.03 |
| | | Dialogue | 0% | 1.00 | 1.00 | undefined | undefined |
| | | Participatory | 0% | 1.00 | 1.00 | undefined | undefined |
| | Teachers | Deficit | 0% | 1.00 | 1.00 | undefined | undefined |
| | | Dialogue | 0% | 1.00 | 1.00 | undefined | undefined |
| | | Participatory | 0% | 1.00 | 1.00 | undefined | undefined |
| | General Public | Deficit | 20% | 0.75 | 0.56 | 0.46 | 0.44 |
| | | Dialogue | 0% | 0.98 | 0.97 | 0.00 | -0.01 |
| | | Participatory | 0% | 0.96 | 0.96 | 0.00 | -0.01 |
| Hands-on | K-12 Students | Deficit | 0% | 1.00 | 1.00 | undefined | undefined |
| | | Dialogue | 5% | 0.86 | 0.83 | 0.36 | 0.35 |
| | | Participatory | 0% | 0.98 | 0.97 | -0.01 | -0.01 |
| | Teachers | Deficit | 0% | 0.98 | 0.97 | 0.00 | -0.01 |
| | | Dialogue | 0% | 0.98 | 0.97 | 0.00 | -0.01 |
| | | Participatory | 0% | 0.96 | 0.96 | 0.00 | -0.01 |
| | General Public | Deficit | 1% | 0.90 | 0.89 | 0.15 | 0.15 |
| | | Dialogue | 0% | 0.89 | 0.88 | 0.00 | -0.05 |
| | | Participatory | 0% | 0.88 | 0.86 | -0.02 | -0.06 |
| Minds-on & Immersive | K-12 Students | Deficit | 0% | 0.99 | 0.99 | 0.00 | 0.00 |
| | | Dialogue | 0% | 0.96 | 0.96 | 0.00 | -0.01 |
| | | Participatory | 0% | 0.99 | 0.99 | 0.00 | 0.00 |
| | Teachers | Deficit | 0% | 1.00 | 1.00 | undefined | undefined |
| | | Dialogue | 0% | 1.00 | 1.00 | undefined | undefined |
| | | Participatory | 0% | 1.00 | 1.00 | undefined | undefined |
| | General Public | Deficit | 4% | 0.95 | 0.94 | 0.58 | 0.58 |
| | | Dialogue | 0% | 0.95 | 0.95 | 0.00 | -0.02 |
| | | Participatory | 0% | 0.99 | 0.99 | 0.00 | 0.00 |



**Code availability**

The codebook used for the content analysis component of this research can be found in the supplement link.

**Data availability**

All data is available in the supplement link or Appendix A.

**Supplement link:**

________

**Author contributions**

Co-conceptualization of this research was completed by CO and EF. Data curation/analysis and writing of the paper was
performed by CO. Review, editing, and supervision was undertaken by EF.

**Competing interests**

The authors declare that they have no conflict of interest.

**Ethical statement**

The work performed in this study is original, reflects the authors' understandings, and does not require the involvement of
human research participants.

**Acknowledgements**

This research would not have been possible without the financial support provided by a Social Sciences and Humanities
Research Council Doctoral Fellowship and a Simon Fraser University Graduate Dean's Entrance Scholarship awarded to
Courtney Onstad. Ian Bercovitz is sincerely thanked for his assistance with interpretation and guidance on formatting statistical
data. YuYen Pan is thanked for her assistance in reviewing the categories and the codebook as part of the content analysis
component of this research. Alice Fleerackers is thanked for their assistance on the method of double coding.




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
