# Peer review of "Geoscience Communication: A Content Analysis of Practice in British Columbia, Canada Using Science Communication Models"

_EGUsphere, 2024_

## Author Response (AR1)

**Reviewer 1 Comments:**

a) Reviewer comments
b) Author's response to comments
c) Author's changes to manuscript (when necessary)

1a)     This manuscript presents original research about how science communication models are used in geoscience communication in British Columbia, finding that the deficit model is still dominant and dialogue and participatory approaches are underused, highlighting a gap between theory and practice.

**General comments:**

The title accurately describes the paper's content and the abstract provides a clear and complete summary. The goals, study design and results are communicated with clarity, with all sections well-structured to effectively convey the research findings and facilitate understanding of the study's significance. The methodologies are well-detailed and limitations are transparently acknowledged, providing a clear understanding of the research's scope and findings. Related and referenced works are properly acknowledged and the reference list reflects an extensive literature review.

The paper addresses an important scientific question under the scope of GC, offering a valuable, comprehensive and current review of the topic. As underlined in the manuscript, there are few studies in this area, making this work a much-needed contribution to advancing the field of geoscience communication. Deficit model thinking, which has received considerable attention in science communication research, needs also to be addressed within the context of geoscience.

Although the work has a regional focus, considering trends studied in other scientific fields, I believe that the findings are likely consistent with broader patterns and therefore, the insights from this paper will contribute to global improvements.

b) We thank the reviewer for the positive feedback on this work. We felt hesitant discussing these trends in a global context, considering the limitations, but we are glad that the reviewer agrees that these trends are consistent with broader trends. The pertinent text to this comment is likely lines 550-552. Please advise if you would like the phrasing changed.

**Specific comments:**

2a)     In the abstract, it is noted that geoscience communication, though rich in practical perspectives, lacks theoretical grounding; it might be beneficial to reinforce this context by including some few more relevant references in the literature review.
b) We agree that this point could be further strengthened by additional references.
c) Section 2.3 (Lines 160-179) was added which includes additional practical perspectives in GC and highlights the few papers that do ground their findings in theory (Illingworth, Stewart & Nield).

3a)     Considering the journal's audience, it may be pertinent to include a few specific examples illustrating each of the three theoretical models of geoscience communication in section 2.2 Science Communication. In the supplemental material, the objectives and activities within the models are

thoroughly explained, however, I would suggest incorporating just a few examples directly into the main text for better context.

b)  We will add a few examples directly in the text.

c) Section 2.3 now incorporates examples of each of the models, and those that are pertinent to geoscience communication.

4a)        In line 84, it is noted that there is often an oversimplification of the diverse and intricate nature of audiences. It is unfortunate that the 'general public' could not be segmented further. Is it possible to provide any insight into who is included in the 'general public'? Were there no activities identified for local communities or media/journalists.

b) This was a particularly challenging decision made to maintain the scope of the study. We observed that websites often did not explicitly state their target audience. While there were examples for local communities, this was less common for media or journalists. Ultimately, our decision was guided by several considerations:

1. **Codebook accessibility**: We aimed to keep the codebook straightforward and manageable for second coders.

2. **Coding complexity**: Including an additional audience category would have introduced 27 new coding options (3 models × 9 resources), further complicating the coding process for second coders.

3. **Statistical limitations**: From a statistical perspective, adding more categories would have exacerbated the challenge of achieving statistical significance. Although statistical analysis was eventually deemed unsuitable due to other issues, this concern influenced our initial decision.

4. **Research objectives**: Considering the overarching goals of the study, we determined that including additional audience categories would not substantially impact the broad trends discussed.

We acknowledge that further clarification on what constitutes the 'general public' audience would be helpful. Please let us know if these adjustments address your concerns

c) Line 282 added (a broad, non-specialist audience or instances where the audience is unspecified)

5a)        It is indeed a pity that geo-art was excluded from the analysis, but this decision is totally understandable due to the limited available information on the objectives.

b) No action required.

6a)        I am not sure if the concept of 'programs for science learning' implicitly emphasizes knowledge rather than engagement.

b) The term in question does originate from an 'informal education' context, which does emphasize knowledge among other goals. However, we felt it was appropriate to use in this context due to its broader informal application, which extends beyond solely knowledge-based objectives. Importantly, this term was used only to categorize activities, and the activities included under this category (as

outlined in the codebook) were not limited to those with a knowledge focus. We hope this clarification helps, but please let us know if further explanation is needed.

c) Emphasized that these terms, including programs for science learning, came from an informal education context (Lines 253-255).

7a)      Regarding the methodology of content analysis (3.2), the description is thorough, reflecting the complexity of the process and demonstrating robustness. The challenges of coding appear to have been significant. In fact, understanding a strategy that looks dialogical at first glance but ultimately turns out to be one-directional in practice must have been quite complex. In that sense, I believe the methodology section thoroughly addresses reliability, which is fundamental for ensuring the validity of the findings and the overall transparency of the research. However, I would consider streamlining that section to facilitate readability and make it easier for the reader to follow without compromising the depth of information.

b) This was also of concern to the authors and will streamline this section in the next iteration.

c) Considerable changes (deleting of the least relevant content) were made to Section 3.2.

8a)      Does this more than twofold discrepancy between participatory objectives (13%) and activities (5%) call for specific further reflection?

b- The following points address this discrepancy.

1)  "The data shows a misalignment between practitioners' stated objectives and the activities they design to achieve them. Deficit and participatory objectives do not consistently translate into deficit and participatory activities. "
2)  "This may be attributable to several factors (e.g. the target audience, communicators' predispositions). "
3)  "Regarding the overrepresentation of participatory objectives relative to activities, this could be due to 1) the comprehensive nature of many participatory activities, allowing a single activity to meet multiple participatory objectives, or 2) a belief that deficit and dialogue activities can still achieve these objectives."
4)  "Beyond the factors mentioned above, it is also possible that practitioners are unaware of the full range of objectives they could aim to address, or have not considered the alignment of their objectives and activities. While we highlight these misalignments, we do not necessarily view them as problematic, especially when more participatory activities are used for less participatory objectives. However, practitioners may be selling themselves short in this case. Conversely, using less participatory activities to meet more participatory objectives may present more significant issues. For example, it is unlikely that lecturing (deficit activity) would allow practitioners to gain the public's involvement in democratic policymaking (participatory objective). "

c) The points above along with a few other key additions (Lines 606-614) have now been grouped under a new heading (Section 5.2) to clarify.

9a)      Can some concrete future steps be suggested to address this deficit model prevalence?

b) This will be further expanded on in the next iteration.

c) A new section (5.4 "Moving Forward") has been added to the discussion which emphasizes scicomm training grounded in science communication theory as a concrete step among others.

10a)    I am not sure if the term 'K-12', is widely recognized by readers globally. Although it is explained in detail in the supplemental material, I recommend providing a small definition in the main text.

b) Agreed.

c) Changed to "K-12" (Kindergarten-Grade 12) in Line 208.

11a)    I would also suggest clarifying the term 'sciart', referring to the intersection of science and art, as it may not be familiar to all readers.

b) Agreed

c) Ended up removing this to consolidate section 3.2.

12a)    Please note that the official designation is 'UNESCO Global Geopark' (UGGp), but the text inconsistently refers to it as 'GeoPark' or 'UNESCO Geopark'.

b) Will change all to UNESCO Global Geopark

c) Changes made throughout.

13a)    Regarding venues, it is not totally clear if Tumbler Ridge UGGp in British Columbia was considered in the assessment?

b) Yes, it was – From Section 3.1: "'Parks', including UNESCO Global Geoparks and national/provincial parks, were considered distinct from natural physical sites due to their managed nature and human-enhanced elements."

c) I've now included all the venues with quotes – maybe this helps clarify?

14a)    **Conclusion:**

Overall, this manuscript is very interesting and well written, offering an important insight into geoscience communication research and contributing to the advancement of the field. The methodology is robust and clearly presented, with results that strongly support the conclusions.

The theoretical studies in this area are notably scarce, making this contribution very valuable. This work presents results supported by concrete data that are essential for developing practical approaches to contribute to address challenges in geoscience communication.

Thanks to the authors for submitting this interesting study.

b) Thank you for recognizing the importance of this work. Your comments have greatly strengthened this paper!

**Reviewer 2 Comments:**

> d) Reviewer comments
> e) Author's response to comments
> f) Author's changes to manuscript (when necessary)

**1a)** **General comments:**

Overall quality: the current study is well-structured, and presents novel data in a clear and interesting fashion, filling an existing research gap in the field. The prominent use of the scicomm theoretical framework as an analysis tool is carried out thoroughly and following a clear background as to why this is appropriate.

Integration of scicomm theory in GC studies.

• Relevance of scientific questions within the scope of Geoscience Communication (GC): scientific question is relevant to field.

• Presentation of novel concepts, ideas, tools, or data: concepts and tools are well-established, data is novel.

• Validity and clarity of scientific methods and assumptions: Clear and valid.

• Sufficiency of results to support the interpretations and conclusions: sufficient.

• Proper credit given to related work and and clear indication of their own new/original contribution: YES

• Title clearly reflect the contents of the paper: YES

• Abstract provides a concise and complete summary: YES

• Structure and clarity of overall presentation: Well-structured and clear

• Fluency and precision of language: Fluent and precise

• Appropriate number and quality of references: YES

b) We thank the reviewer for the positive feedback on this work.

 Additional comments:

1a)        A general comment on the literature review: I would suggest shortening the chapter on scicomm and including a third chapter of the lit. review outlining commonalities between GC and scicomm. Although the parallel fields of GC and scicomm are presented clearly, the review could provide a stronger basis supporting your claim that integration of these fields is pertinent.  Perhaps in continuation to the relationship between scicomm models and objectives (as per Metcalfe) that you have provided, you could also highlight overlaps with GC objectives, thus providing a basis for the claim that the scicomm models are relevant to GC. Another option could be to include GC studies that integrated scicomm models (if these exist).

b) We agree with this suggestion. Integration of both fields is a great idea and there are many opportunities to discuss the overlap in objectives and activities.

c) « Chapter 2.3 (Geo)Science Communication » has been added to the literature review, and Chapter 2.2 has been shortened. Chapter 2.3 now highlights the commonalities between the two and includes examples. We have also further emphasized GC and scicomm parallels in the Introduction to strengthen the rationale for the study.

2a)      A general comment regarding research questions: the research question is currently "hidden" within chapter 3.2 content analysis. Consider including the research question in a more prominent and perhaps earlier section of the paper (perhaps in the first paragraph of the Methods chapter).

b) Agreed. Will address this in the next iteration.

c) Research questions have been included and explicitly stated in the Introduction Section. (Lines 39-46)

3a)      General comment on the discussion: please make sure that the reader is clear on which conclusions are an outcome of this study including phrases such as "we found" or "in this study we discovered". Please compare your own findings with those of other studies, in the field of GC if available, and if not in other fields. Below I have provided some examples of where this should be included, but there are additional instances.

b) This is helpful advice, and we will be sure to review the discussion section and make these changes. While I have tried to compare our results to as many relevant studies as possible, I believe some of these findings are novel, as they have not been previously discussed in the context of science communication models. For instance, I was unable to compare the finding that dialogue activities have increased through virtual workshops to any existing science communication studies, as the available literature largely focuses on examples from formal education rather than science communication contexts.

c) Additional phrases clarifying the nature of findings have been incorporated throughout the discussion section. Specifically, the discussion now relates findings to the following topics: (1) the prevalence of the deficit model, (2) hands-on activities as a method for engaging youth, (3) natural physical sites as venues for place-based participatory engagement and citizen science, and (4) the underrepresentation of the participatory model in training initiatives.

4a)      Discussion: perhaps there should be three subsections here? (1) The prevalence of the deficit model in GC communication in B.C. (2) Why the deficit model persists (3) Going past the deficit model. I think that this last subsection is of interest and could be developed further.

 b) We agree and will make these changes.

c) We ended up including four subsections of the Discussion: 1) Deficit Model Persists – talks about general results of all three models which shows that the deficit model is most prevalent, 2) Misaligned objectives and activities, 3) Why does the deficit model persist?, 4) Moving forward

**Specific comments**

1a)   Lines 26-27 - there is mention of absence of "robust theoretical frameworks in the geoscience communication literature". I am not sure whether you are referring to tailored GC frameworks, or the integration of existing science communication theory into GC studies, as demonstrated in the current study. If the former, I suggest including a succinct explanation on the unique qualities of GC that warrant a tailored framework. See for example: https://oxfordre.com/communication/display/10.1093/acrefore/9780190228613.001.0001/acrefore-9780190228613-e-311?p=emailAopIlgda1dpUk&d=/10.1093/acrefore/9780190228613.001.0001/acrefore-9780190228613-e-311 where there is clarity on what makes explanation-as-teaching unique to other forms of risk messaging. If the latter, possibly rephrase. Additionally, consider adding more recent citations to strengthen the assertion that there are no robust theoretical frameworks in GC literature to date (last citation is dated 2017).

b) We will address this.

c) We have now more clearly specified this in the introduction. However, we do also mention that there are unique aspects to geoscience comm and provide a few examples. We also added more recent citations (Ganie et al., 2024, Illingworth 2018, Stewart & Lewis, 2017) in geoscience contexts, and also noted that other scientific disciplines have adopted this approach of utilizing scicomm theory (Lines 34-36).

2a)   In the abstract it is stated that: "The study highlights disparities in the use of communication models across *target audiences*, regions, and venues". However in lines 45-46, the audiences are not referenced: "Additionally, the geographic locations and venues where geoscience communication occurs will be examined to assess the prevalence of the various models". I suggest rephrasing for clarity as well as mentioning the examination of target audiences as well.

b) This appears to be an oversight. Will add this.

c) This ended up being removed from the text, but I have ensured that audiences are included throughout.

3a)   2.1 GC - I think that it would be a good idea to begin with some discussion on aims and the potential of reaching them. Although beginning with two aims (raising awareness & stimulating discourse), the chapter continues with two additional aims (addressing society's most pressing challenges and enhancing geo-literacy). I am unclear as to issues such as: (1) which of these aims are being reached and which are not; (2) whether GC goals and practice are aligned?; (3) What the central "practitioners' real-world challenges" mentioned in the introduction are? Possibly by addressing these issues & including research gaps, this chapter could be a basis for the stated need to analyze whether GC has transitioned from deficit to dialogue approaches. At the moment it is unclear to me why this analysis is pertinent.

b) The aims discussed at the beginning of the paper were intended as a definition of geoscience communication rather than an emphasis on specific objectives. Similarly, the mention of the "two additional aims" was meant to highlight some pressing issues that geoscience communicators are addressing, not necessarily to frame them as model-oriented objectives.

Regarding comment 1, are you asking whether geoscience communication practitioners are meeting these objectives? If so, I'm unsure of the necessity of including such findings, particularly when balancing this with the overall length of the paper.

For comment 2, I have not come across research that examines whether geoscience communication goals and activities are aligned, which is partly why we aimed to address this in our study.

For comment 3, the referenced section was ultimately removed.

Finally, I wonder if part of the issue stemmed from the placement of the research questions, which were previously located in Chapter 3. With their more prominent placement now, does this address your concern? Please let me know if we have misunderstood your comments on this point.

c) We revised the Introduction to better emphasize the gaps in the literature and added Chapter 2.3 to more explicitly align the science communication models with geoscience communication practices. For example, in lines 161–165, we highlight that the deficit model currently appears to be commonly used in geoscience communication. Additionally, in lines 58–73, we have more clearly and explicitly articulated the importance of our research questions. We hope these changes more intuitively demonstrate the relevance and significance of our analysis.

4a)          Lines 131-134: Above it is claimed that information retrieval was systematic. Therefore, perhaps clarify the combinations between keywords used. Minimum of two keyword combinations? Three? Were all possible combinations between the keywords entered? How did you choose the order of keywords for your search? Additionally, since the timeline for the search is relatively long, perhaps clarify whether the searches were carried out multiple times during the time-period or was each combination entered once? Did you look at all the results received from each Google search?

b) In complete honesty, the database compilation component of this research was not done to the best standards (this was the first couple months of my PhD). The word systematic was intended to describe the content analysis, thus we will delete this and instead include that term in the content analysis section. However, there are a few additional details we could include, as you have suggested.

c) Added line 193-195 and deleted the word systematic from the methods section but included it in the content analysis section.

5a)          Database - was this provided in the supplementary materials? If not, please consider including it together with a reference to the DB within the text, or at least the subset analysed in the current study.

b) Not it was not, but we will add this.

c) Provided with new iteration.

6a)          Table 2: please explain the "prevalence" column. Unclear what is shown in this column. In addition, please provide reference for "marginals" in this table. Are these ratings that were neither clearly present nor absent?

b) Will add these details.

c) Line 378 was added.

7a)      Line 337: I think for clarity it would be helpful to begin with "The full sample encompasses a total of X objectives and Y activities" and then continue with the breakdown: "including the following: A. agreements between coders 1 and 2 (596 for objectives, 6137 for activities…" etc.

b) Agreed.

c) Added line 420.

8a)      Throughout the results: please include numbers together with the percentages (as shown in the figures).  E.g.: "an overwhelming 76% (n=118) were deficit…"

b) Will correct this.

c) This adjustment was made throughout the results section.

9a)      Line 343 (Results): The number 155 appearing in "Out of the 155 geoscience communication objectives" seems to appear quite suddenly for the reader. I am not sure what this number represents. Is this the total number of objectives that were coded? Or are these objective categories that were created following coding? The same goes for the number 363 (activities) which has simply appeared here. I suggest detailing the total number of objectives and activities found and coded, followed by the total objective categories and activity categories (if these exist). This information could appear in the methodology or results.

b) Will address this.

c) With line 420 added, this now shouldn't be a shock.

10a)      Lines 351-354: first, I would recommend a subtitle here (e.g. models by audience and medium). Second, there is no reference to fig 3a within the results. I think it should be appearing somewhere in these lines as there is data here that does not reference a figure/ table but seems to be represented in fig 3a.

b) These results were intentionally excluded under a specific subtitle since they represent the overall results (solely the objectives and activities without discussion of the region, resources, or venues). Audiences are discussed throughout all sections because they were inherently included as part of the coding process and notably have been verified by a second coder. Could include a subtitle of "General results"? Please advise.

c) Reference to Fig3a has been added. No change made to subtitle.

Fig 3: First, please clarify what the numbers within the bars (3a & 3b) are. If they are percentages, note that they are different from the numbers appearing within the text. Second, note that you have referenced use of colors which will need to be rephrased since colors will not appear in the final manuscript (as per editor comment).

b) The numbers correspond to the frequencies (thus are the same values as the first graphs). Will fix colours. I believe the editors comment on colour was solely for Tables – figures are allowed to use colour.

c) Have specified "n=" in the graphs. Kept colour in graphs but changed figure 3c by showing the groupings with grouping brackets as opposed to another layer of colours.

11a) Another issue I am not entirely clear about is the analysis of models. It appears that sometimes you have analyzed both *activities* and *objectives*, sometimes only activities, for example in the case of resources for example, as the title "Model *Activities* in Resources" suggests. Why have the objectives not been included in the analysis in the case of resources? I am unclear for the reason for this discrepancy (perhaps you explained it in the methodology but I could not find an explanation), please clarify it in the results chapter.

b) Objectives and activities are treated separately as shown in concept map. It appears that I only discuss objectives in the recording/coding section, so will add details on this.

c) Added line 244 and 247-248 to further clarify this.

12a) Fig 4: titles for y axis are missing.

b) Will correct.

c) line 503: Added "frequency" on the y-axis to the far-left graph and added "Frequencies (y-axis)" to the figure caption.

13a) Line 421: clarification that B.C. refers to British Canada could be important for non-Canadians reading this paper.

b) Will fix.

c) Changed all to "British Columbia".

14a) Lines 459-463: I have discussed the following statement: *"The data shows a misalignment between practitioners' stated objectives and the activities they design to achieve them. Deficit and participatory objectives do not consistently translate into deficit and participatory activities"* as well as the example you have used to depict misalignment, with a scicomm expert with a strong background in education. I would suggest consulting with additional experts in education/ scicomm, as this statement may diverge from scicomm and education theory. The statement and example provided suggest that there is a pedagogical misalignment between the use of hands-on activities to promote education. However, hands-on activities are an established and pedagogically sound method to promote education according to both theory and practice, as you state correctly in lines 475-476. Therefore, could it be possible that you have incorrectly classified all hands-on activity as dialogical? The expert I consulted with suggested that some hands-on activities (e.g. cooking class) could be classified under dissemination. I would suggest double-checking your research methods here, and ensuring that the conclusion you have reached is not an outcome of activity misclassification.

b) We consulted with a formal education expert but were unable to reach a consensus on the intended meaning of this comment. Regarding the potential miscoding of hands-on activities, we adhered strictly to the codebook instructions. The following terms, all associated with the dialogue model, were most frequently linked to hands-on activities: 'activity involving people in science,' 'provide access to scientists,' 'train/develop skills to participate in science,' and 'workshops.' Both coders, an external expert, and the formal educator agreed on these classifications. However, we chose not to explicitly argue this in the paper, as that would require justifying all coding decisions, which would be beyond the paper's scope.

While we acknowledge that deficit model objectives were often associated with these activities, our focus was solely on coding the activity itself. However, the objectives (for practitioners who had hands-on activities) were often coded as deficit model objectives, resulting in the misalignment. This is the misalignment we refer to in Section 5.2, which we have now made more explicit. We hope this clarifies the issue. Additionally, we would like to reiterate that our study focused exclusively on informal education, not formal education, which might explain some of the confusion. Please let us know if the revisions and this clarification address your concerns.

c) Rephrasing of misalignments in Lines 598.

15a)    Line 471: please provide examples or justification for the conclusion: "However, practitioners may be selling themselves short in this case."

b) Agreed.

c) Ended up deleting this.

16a)    Lines 472-473: please provide reference for the example.

b) Will fix.

c) Added Metcalfe, 2019b reference to line 604.

17a)    Line 474: Should begin with "*here we found that in B.C.* the deficit model is used most when communicating with general public audiences, and least with K-12 students".

b) Agreed.

c) Rephrased in Lines 553-556.

18a)    Line 475: "along with the prevalence of the dialogue model when communicating with K-12 students" - do you mean in this study? Is that not essentially the same finding as the one above according to which the deficit model is used least among K-12 students? The wording makes it sound like there are two separate findings backing your conclusion.

b) Will rephrase.

c) Rephrased in Lines 553-556.

19a)    Lines 475-476: "the notion that handson activities are particularly effective in engaging youth in an educational context" - please reference this statement.

b) Will add a reference.

c) Added Kyere, 2017 ref to Line 556.

20a)    Lines 476-478: "Notably, participatory activities, such as citizen science, are more prevalent among general public audiences, while educators appear to have limited access to such activities". Please elaborate: in this study? In general in Canada? Please compare with the findings in other studies and fields.

b) Will revise.

c) Specified "in this study" to Line 556. Found a new reference mirroring my exact findings – added Giardullo et al., 2023 to line 559.

21a)    Lines 484-485: according to this sentence, with refers to the previous one, I understand that citizen science is dialogical, please rephrase.

b) Will rephrase.

c) Added "respectively" to Line 568.

22a)    Line 501: Regarding the primary geographic locations of services offered, it's *noteworthy* that targeted participatory activities are *notably* lacking. Rephrase: no need for both "noteworthy" and "notably".

b) Will rephrase.

c) Deleted.

23a)    Line 550: I suggest elaborating on what is being co-designed in the co-design process, in case not all readers are familiar with this term.

b) Will further elaborate on this concept.
c) Added "where diverse stakeholders work together with equal power and say in an outcome" to line 721-722.

**Technical corrections**

Line 69: closing parentheses missing: (Blackwood, 2009; Royal Ontario Museum, 2021**)**

b) Will fix.

c) Fixed.

---

## Referee Report (RR1)

This paper is a clear improvement over the previous version submitted earlier. The authors have addressed all the concerns raised, providing clear explanations and necessary clarifications. The revisions made have enhanced the overall clarity, coherence and quality of the manuscript. I appreciate the effort the authors have put into refining the work and I believe the manuscript is now in a much-improved state. I am sure this will be a major contribution to the field.

---

## Author Response (AR2)

a) Reviewer Comments
b) Author's response to comments
c) Author's changes to manuscript

*I suggest two minor amendments:*

*1.*

*a) Lines 32 - 39 regarding the research questions: should be moved after line 51, as otherwise, the terminology used within the questions has not yet been introduced to the reader.*

b) Agreed.

c) Rearranged in new version.

*2.*

*a) Lines 478 - 480: "We observed a significant misalignment when K-12 students were the target audience, in that hands-on activities (coded as a dialogue activity) were often used to educate (coded as a deficit objective)". As I noted earlier, in the field of education, such activities are considered a pedagogically sound method of education, therefore seems to be problematic to claim that there is a misalignment between "hands-on activities" (dialogical) and the goal of education (dissemination). Perhaps this sentence should therefore be rephrased.*

b) I get the feeling that the reviewer and I are agreeing, but we are taking on different meanings of the term "misalignment". I have added two clarifying sentences but have not changed the original sentence since that sentence follows the same structure I used throughout this paragraph.

c) Added the following after this sentence: "While using hands-on activities to meet educational goals is a well-known pedagogical practice in formal education, the science communication models are conceptually distinct, especially when applied prescriptively through a content analysis approach. This constraint is discussed further in the Limitations section."